# Genetic Restoration of Heme Oxygenase-1 Expression Protects from Type 1 Diabetes in NOD Mice

**DOI:** 10.3390/ijms20071676

**Published:** 2019-04-03

**Authors:** Julien Pogu, Sotiria Tzima, Georges Kollias, Ignacio Anegon, Philippe Blancou, Thomas Simon

**Affiliations:** 1Centre de Recherche en Transplantation et Immunologie, Institut National de la Santé Et de la Recherche Médicale (INSERM), Université de Nantes, 44000 Nantes, France; pogujulien@gmail.com (J.P.); Ignacio.Anegon@univ-nantes.fr (I.A.); blancou@unice.fr (P.B.); 2Institute of Immunology, Biomedical Sciences Research Centre “Alexander Fleming”, Vari, 210 Attica, Greece; sotiria.tzima@roche.com (S.T.); kollias@fleming.gr (G.K.); 3Université Côte d’Azur, Centre National de la Recherche Scientifique (CNRS), Institut de Pharmacologie Moléculaire et Cellulaire, 06560 Valbonne, France

**Keywords:** ANTIGEN presenting cell, tolerance, Tet-ON system, antigen presentation

## Abstract

Antigen-presenting cells (APCs) including dendritic cells (DCs) play a critical role in the development of autoimmune diseases by presenting self-antigen to T-cells. Different signals modulate the ability of APCs to activate or tolerize autoreactive T-cells. Since the expression of heme oxygenase-1 (HO-1) by APCs has been associated with the tolerization of autoreactive T-cells, we hypothesized that HO-1 expression might be altered in APCs from autoimmune-prone non-obese diabetic (NOD) mice. We found that, compared to control mice, NOD mice exhibited a lower percentage of HO-1-expressing cells among the splenic DCs, suggesting an impairment of their tolerogenic functions. To investigate whether restored expression of HO-1 in APCs could alter the development of diabetes in NOD mice, we generated a transgenic mouse strain in which HO-1 expression can be specifically induced in DCs using a tetracycline-controlled transcriptional activation system. Mice in which HO-1 expression was induced in DCs exhibited a lower Type 1 Diabetes (T1D) incidence and a reduced insulitis compared to non-induced mice. Upregulation of HO-1 in DCs also prevented further increase of glycemia in recently diabetic NOD mice. Altogether, our data demonstrated the potential of induction of HO-1 expression in DCs as a preventative treatment, and potential as a curative approach for T1D.

## 1. Introduction

Heme oxygenase-1 (HO-1) is one of the three isoforms of the heme oxygenase enzymes that catabolizes the degradation of heme into biliverdin, free iron, and carbon monoxide (CO). In contrast to the two other isoforms, HO-1 is the only one which is induced by oxidative stress or pro-inflammatory cytokines. Under stressful conditions such as increased level of reactive oxygen species (ROS), the nuclear factor erythroid 2-related factor 2 (Nrf2) is released from the inhibitor protein Kelch-like ECH-associated protein 1 (Keap1). After translocation in the nucleus, Nrf2 binds to the antioxidant response element (ARE) sequence and increases the transcription of anti-oxidant genes including HO-1 (*Hmox-1* ) and NAD(P)H quinone dehydrogenase 1 (*nqo-1*) [1]. Heme oxygenase-1 is also induced by its own substrates, whether they are natural (heme or hemin) or synthetic (Cobalt-protoporphyrin, CoPP). Many lines of evidence have pointed towards an anti-inflammatory role for HO-1. A chronic inflammation state was evidenced in both HO-1 knock-out mice [2] and in a patient exhibiting a mutation in the HO-1 coding *Hmox-1* gene [3]. Induction of experimental autoimmune encephalomyelitis (EAE) in HO-1 knock-out mice led to enhanced neurological symptoms as compared to wild-type (wt) mice [4]. In agreement with this latter result, both genetic and pharmacological manipulations aimed at inducing HO-1 expression protected mice against EAE [4], autoimmune type 1 diabetes (T1D) [5,6], and allergic asthma [7]. Furthermore, we and others have shown that HO-1 and CO treatment improved graft survival in both mice and rats [8,9,10,11,12]. Although the anti-inflammatory properties of HO-1 rely on heme degradation products, i.e., biliverdin, free iron, and CO, the cellular targets of these latter molecules remain to be identified. In this respect, studies in both mice and humans have ruled out the possibility that HO-1 degradation products act directly on regulatory T-cells (Tregs) [13,14,15]. In contrast, other studies have suggested that HO-1 degradation products could promote the development of tolerogenic dendritic cells (DCs) [16,17,18], that could eventually inhibit T-cell responses either by inhibiting the migration of autoreactive CD8+ T-cells to the target organ [19] or by inducing Tregs cells [20].

Type I diabetes (T1D) is a chronic autoimmune disease that results from the killing of pancreatic β-cells by autoreactive T-lymphocytes. Clinical diagnosis is preceded by a prediabetic asymptomatic phase characterized by the infiltration of several immune cell types including CD4+ and CD8+ T-cells in pancreatic islets. The diabetes-prone non-obese diabetic (NOD) mouse strain has been extensively studied as a clinically relevant model of T1D [21]. Non-obese diabetic mice, and particularly females, spontaneously develop insulitis starting at 4–5 weeks of age. A significant proportion of these animals (between 20 and 80% depending on housing conditions) eventually progress to diabetes as the result of pancreatic β-cells destruction by CD4^+^ and CD8^+^ T-cells. Dendritic cells (DCs) play a critical role in the initiation of TD1 by capturing and processing β-cells antigens. These DCs then migrate to the pancreatic lymph node where they present antigenic peptides to diabetogenic naïve T-cells leading to their activation and differentiation into effector T-cells [22].

Given the anti-inflammatory properties of HO-1 and the critical role of DCs in the initiation of T1D, we hypothesized 1) that DCs from NOD mice could be deficient in HO-1 expression, and ii) that the selective upregulation of HO-1 in DCs could inhibit the development of T1D. Here we demonstrated for the first time that genetic induction of HO-1 limited to DCs was sufficient both to prevent T1D in non-diabetic NOD mice and to stabilize glycemia in some recently diabetic NOD mice.

## 2. Results

### 2.1. DCs from NOD Mice are Deficient in HO-1 Expression

In contrast to NOD mice that spontaneously develop insulitis and eventually T1D, Major Histocompatibility Complex (MHC)-matched non-obese diabetes resistant (NOR) mice only developed peri-insulitis and never progressed to insulitis and T1D [23]. We found that the percentage of HO-1-expressing cells among spleen CD11c^+^ cells was two-fold lower in NOD mice compared to NOR mice (1.8% versus 3.8%) demonstrating a strain-specific difference (Figure 1A). To investigate whether the lower percentage of HO-1-positive DCs in NOD mice resulted from an intrinsic defect, we generated bone marrow-derived DCs (BMDCs) from both NOD and NOR mice and assessed these cells for HO-1 expression before and after exposure to the HO-1 inducer CoPP. In the absence of CoPP, the percentage of HO-1-positive DCs was lower in BMDCs from NOD mice compared to those from NOR mice. While CoPP increased the percentage of BMDCs expressing HO-1 in both NOD and NOR mice in a dose-dependent manner, the percentage of HO-1-positive BMDCs remained lower in NOD mice whatever the concentration of CoPP (Figure 1B). Altogether, these data demonstrated that DCs from NOD mice exhibited an intrinsic defect in HO-1 expression.

### 2.2. Generation of a HO-1 Inducible Transgenic Mouse Strain on the NOD Background

To investigate whether the upregulation of HO-1 in DCs could impact the development of T1D in NOD mice, we used a pIi-tTA^+^ transgenic mouse strain in which the tetracycline transactivator (*tTA*) gene was under the control of the MHC-II invariant chain (Eα-Ii) promoter [16]. We generated a new transgenic mouse strain, termed TetO-HO-1^+^, in which the *Homx-1* gene was cloned downstream a hybrid cytomegalovirus-Tet operator (Figure 2A). We then crossed pIi-tTA^+^ to TetO-HO-1^+^ transgenic mice, identified double transgenic animals, and further crossed these mice onto the NOD background for eight generations. We expected the resultant TetO-HO-1^+^ pIi-tTA^+^ double-transgenic mice to show a doxycycline (DOX)-driven expression of HO-1 in MHC-II^+^ cells. To investigate whether this was the case, we purified splenic CD11c^+^ from double transgenic TetO-HO-1^+^ pIi-tTA^+^ and single transgenic TetO-HO-1^+^ pIi-tTA^-^ mice and incubated these cells with different doses of DOX (Figure 2B). As expected, DOX did not increase the percentage of HO-1-positive DCs in single transgenic TetO-HO-1^+^ pIi-tTA^-^ mice, but it did in TetO-HO-1^+^ pIi-tTA^+^ double transgenic mice. An increase in the percentage of HO-1-positive DCs was observed as early as two hours after DOX treatment in a dose- and time-dependent manner.

To investigate whether DOX could upregulate HO-1 expression in vivo and to further identify in which cell types we treated or not TetO-HO-1^+^ pIi-tTA^+^ double-transgenic mice with DOX for 24 h, we then analyzed splenic cells for HO-1 expression by flow cytometry after intercellular staining with anti-HO-1 monoclonal antibody (mAb) and surface staining with mAbs directed to the identify APCs (CD11c to identify DCs; B220 to identify B cells; F4/80 and CD11b to identify macrophages, Appendix A). While DOX increased the percentage of DCs that expressed HO-1, it did not have any impact on the percentage of macrophages or B lymphocytes expressing HO-1 (Figure 3A). As expected, DOX increased the percentage of HO-1-expressing DCs TetO-HO-1^+^ pIi-tTA^+^ double-transgenic mice, but not in their TetO-HO-1^+^ pIi-tTA^−^ single transgenic littermates (Figure 3B). Of note, the percentage of HO-1-positive DCs in DOX-treated TetO-HO-1^+^ pIi-tTA^+^ double transgenic mice was comparable to the percentage of HO-1-positive DCs in NOR mice.

### 2.3. Upregulation of HO-1 in DCs Prevents T1D in NOD Mice

We next investigated whether the selective upregulation of HO-1 in DCs could prevent T1D in NOD mice. We treated TetO-HO-1^+^ pIi-tTA^+^ double transgenic mice and their TetO-HO-1^+^ pIi-tTA^−^ single transgenic littermates with DOX starting at 4 weeks of age. Control single and double transgenic mice were left untreated. Doxycycline treatment did not have any impact on the incidence of T1D in TetO-HO-1^+^ pIi-tTA^−^ single transgenic mice (Figure 4A). Likewise, TetO-HO-1^+^ pIi-tTA^−^ single transgenic mice and TetO-HO-1^+^ pIi-tTA^+^ double transgenic mice developed T1D with a similar incidence in the absence of DOX treatment. In striking contrast, only 10% of DOX-treated TetO-HO-1^+^ pIi-tTA^+^ double transgenic mice developed T1D before 9 months of age, compared to 70% of DOX-treated TetO-HO-1^+^ pIi-tTA^−^ single transgenic mice (Figure 4A). In agreement with this latter result, the percentage of islets exhibiting immune cell infiltration was strongly reduced in DOX-treated TetO-HO-1^+^ pIi-tTA^+^ double transgenic mice compared to single transgenic controls (Figure 4B).

### 2.4. Upregulation of HO-1 in DCs Prevent A Further Increase in Glycemia in Recently Diabetic NOD Mice

We next investigated whether the selective upregulation of HO-1 in DCs could impact glycemia in recently diabetic NOD mice. To this aim, we monitored glycemia in TetO-HO-1^+^ pIi-tTA^+^ double transgenic mice and TetO-HO-1^+^ pIi-tTA^−^ single transgenic mice every day. Mice exhibiting a glycemia higher than 200 mg/dL for two consecutive days were immediately given DOX in their drinking water and further followed for glycemia. While all animals in the TetO-HO-1^+^ pIi-tTA^−^ single transgenic mice group had to be sacrificed because they reached humane end-points, three animals out of eight in the TetO-HO-1^+^ pIi-tTA^+^ double transgenic mice group remained normoglycemic. Glycemia progressively increased in TetO-HO-1^+^ pIi-tTA^−^ single transgenic mice following DOX treatment, whereas it remained constant in DOX-treated TetO-HO-1^+^ pIi-tTA^+^ double transgenic mice (Figure 5). Therefore, the selective upregulation of HO-1 in NOD DCs could inhibit T1D progression in recently diabetic animals.

## 3. Discussion

Type 1 diabetes is a complex multifactorial autoimmune disease in which genetic factors play a critical role. As many as 60 genetic susceptibility loci, termed insulin-dependent diabetes (Idd) loci, have been identified in both mice and humans. Here, we have shown that DCs from NOD mice are deficient in HO-1 expression compared to those from NOR mice that do not develop T1D, although these two strains share 88% of their genome [23]. To which extent this defect contributes to the susceptibility of NOD mice to T1D remains a matter of speculation. While in humans the gene coding for HO-1, *Hmox1*, does not map to any identified Insulin dependent diabetes (Idd) locus, the gene coding interleukin-10 (IL-10), which regulates HO-1 expression [25], colocalizes with Idd3 [26]. It has been previously demonstrated that IL-10 could induce HO-1 in antigen presenting cells (APCs) through Signal transducer and activator of transcription 3 (STAT-3)- and Phosphoinositide 3-kinase (PI3K)-dependent mechanisms [25,27]. Given the fact that decreased IL-10 levels have been associated with T1D in both NOD mice and humans [28,29], this may explain the low level of HO-1 observed in NOD DCs. This defect of HO-1 expression in DCs may also extend to other HO-1 expressing cells including macrophages and B lymphocytes.

On a different but related topic, several defects associated to APCs’ phenotype and functions have been identified in NOD mice that may explain why this strain is prone to T1D. As an example, hyperactivation of nuclear factor kappa-light-chain-enhancer of activated B cells (NF-κB) has been detected in DCs of NOD mice, and was directly correlated to elevated levels of IL-12 secretion relative to DCs from control animals [30]. Elevated NF-κB activation in DCs resulted in an increased capacity of these cells to stimulate naïve CD8^+^ T cells and promote their differentiation into cytotoxic effector T-cells [31]. Decreased expression of molecules associated with tolerance induction (CD103, Langerin, C-type lectin domain family 9, member A (CLEC9A), C-C chemokine receptor type 5 (CCR5) or increased expression of co-stimulatory molecules (CD80, CD86) [32] in the DC population of NOD mice also suggested that abnormal differentiation of pancreatic DCs contributes to the loss of tolerance. Moreover, reduced numbers of tolerogenic DCs have been observed in NOD mice [31]. Altogether, these studies demonstrated that NOD DCs possess intrinsic characteristics that may contribute to the autoimmune phenotype. Whether there is a causal relationship between these defects and HO-1 partial deficiency remains to be investigated.

We have generated transgenic NOD mice in which HO-1 expression is regulated by the Tet ON system in MHC class II-positive cells. Surprisingly, HO-1 induction seems to be restricted to DCs among MHC-II^+^ cells. This could be due to the Tet ON system which was under the control of the MHC-II invariant chain (Eα-Ii) promoter. Indeed, RNA sequencing data from the Immgen Consortium showed that mRNA levels of the MHC-II invariant chain (CD74) was higher in splenic DCs compared to splenic B lymphocytes and macrophages, respectively (Appendix A). Thus, a higher expression of Eα-Ii in DCs could be associated with a higher sensitivity to DOX resulting in HO-1 upregulation in DCs at a similar level as observed in the NOR control strain. Recovery of HO-1 expression in DCs to levels comparable to NOR mice in DOX-treated pIi-tTA-tHO-1 NOD mice dramatically lowered T1D incidence. While the proportion of DCs expressing HO-1 in non-diabetic mice is low, their capacity to induce tolerogenic immune response may be very powerful, even with a small proportion of cells expressing HO-1 as previously shown [19]. Concerning the role of HO-1 negative DCs, we can hypothesize that these cells might be of physiological importance to induce inflammatory response upon detection of immune danger signals.

The low level of protection observed in DOX-treated diabetic NOD mice (3 animals out of 8) may be due to either the limited induction of HO-1 in DCs or the extent of beta cells’ destruction of time of intervention. The efficacy of protection might be improved by combining HO-1 induction in DCs with immunosuppression treatments.

We [19] and others [5] have previously shown the protective effect of upregulating HO-1 in T1D with chemical inducers or systemic HO-1 transduction. Here, we demonstrated for the first time that genetic induction of HO-1 limited to DCs was sufficient both to prevent T1D in non-diabetic NOD mice and to stabilize glycemia in some recently diabetic NOD mice. As an anti-inflammatory protein, HO-1 exerts its immunomodulatory effects through multiple mechanisms by modifying T-cell responses, either directly at the level of the T-cell, or, most likely, by indirectly influencing APCs [33]. HO-1 induction inhibits DCs’ maturation and secretion of pro-inflammatory cytokines [8]. Moreover, HO-1+ DCs or DCs treated by the HO-1 end-product CO, showed impaired antigen presentation abilities [4,34] possibly resulting in fewer islet-specific pathogenic T-cells. The absence of HO-1 expression in APCs also impaired suppressive functions of Tregs [35]. The low proportion of HO-1^+^ DCs in NOD mice could in part explain the Treg defect observed in T1D. We also demonstrated previously that ex vivo treatment with CO conferred tolerogenic properties to DCs and inhibited the pathogenicity of naïve T-cells stimulated by those DCs in a T1D model [17]. Moreover, intradermal injection of HO-1 inducers promoted the accumulation of DCs overexpressing HO-1 in draining lymph nodes (LNs) [19]. These HO-1^high^ DCs exhibited antigen-specific tolerogenic properties as demonstrated by their ability to inhibit the diabetogenic potential of autoreactive cytotoxic T-cells. The same protective mechanisms could also be at play following the restoration of HO-1 level in DCs of DOX-treated pIi-tTA-tHO-1 NOD mice.

Another mechanism of protection could rely on the antioxidant properties of HO-1. As an example, the induction of HO-1 by hemin treatment has been showed to reduce hyperglycemia and to improved glucose metabolism in streptozotocin-treated rats [36] [. The protective effect and the reduction of lesions in the pancreas were due to the inhibition of oxidative stress mediated by HO-1 activity. The oxidative stress mediated by hyperglycemia is a major pathophysiological factor in T1D [37] that could be reduced by the genetic induction of HO-1 limited to DCs in our model.

Altogether, our results showed that restoration of HO-1 expression levels in DCs in NOD mice prevent the development of T1D but also highlight the therapeutic beneficial effect of inducing HO-1 in APCs as a treatment for T1D.

## 4. Materials and Methods

### 4.1. Animals

The NOD/LtJ and NOR/Lt mice were originally purchased from Charles River. The pIi-tTA mice were a kind gift from Christophe Benoist [38]. For generation of the TetO-HO-1 mice, the human α-globin intron located upstream of the cDNA sequence, the human HO-1 cDNA, and the bovine growth hormone polyA located downstream of the cDNA were cloned at the Not-I/Xho-I sites into pBluKSM-tet-O-CMV vector containing the Tet-Responsive-Element (TRE) downstream of the minimal CMV promoter followed by the human α-globin intron and the bovine growth hormone polyA. Transgenic mice were generated by pronuclear microinjection of CBA/C57BL6 eggs with the XhoI-NotI fragment of the vector described. Seventeen different founders were carrying the transgene as tested by PCR and Southern blot. Of the 17 lines, 3 founders contained high copies of the hHO-1 cDNA. One of these was further analyzed by crossing with actin-rtTA mice. The hHO-1 expression was confirmed by Western blotting. Finally, both strains pIi-tTA and TetO-HO-1 were backcrossed to NOD/LtJ mice for at least eight generations. Only females were used in experiments. All animal breeding and experiments were performed under conditions in accordance with the European Union Guidelines.

### 4.2. Genotyping of rTA-HO-1

Mice were bled from the cheek into 100 μL of heparin (125 UI/mL). For DNA purification, we used the DNeasy Blood and Tissue Kit following the manufacturer’s instructions. Consequently, we performed a PCR for each transgene inserted in the transgenic mice: rtTA and TetO-HO-1using GoTaq G2 Flexi DNA Polymerase (Promega, Madison, WI, USA). Then, samples were loaded in an agarose gel electrophoresis. The mice that only had the *TetO-HO-1* gene but lacked the *tTA* gene were named littermates and used as controls.

### 4.3. Doxycycline Treatment

Doxycycline hyclate powder (Sigma–Aldrich, St. Louis, MO, USA) was diluted in drinking water at different concentrations (100 µg/mL up to 800 µg/mL), protected from light, and were changed every 2 to 3 days.

### 4.4. Diabetes Follow-Up

Diabetes monitoring was done by daily glycemia measurement to identify diabetic mice. Mice were considered diabetic when glycemia was superior to 200 mg/dL for two consecutive days. For incidence follow-up, glycosuria was measured twice a week and diabetic animals were confirmed by glycemia measurements (superior to 200 mg/dL for two consecutive days).

### 4.5. Flow Cytometry

Spleen was digested in collagenase D (Roche, Basel, Switzerland) during 30 min at 37 °C then crushed and filtered on 70 µm tamis before staining. Single-cell suspensions were stained with mouse F4.80 (BM8, eBioscience), mouse CD11c (HL3, BD), and mouse B220 (RA3-6B2) antibodies from BD Biosciences. The HO-1 Intracellular stainings were performed using the BD cytofix/cytoperm kit with anti-HO-1 antibody (clone HO-1-1, abcam) followed by anti-mouse IgG1 (clone A85-1) conjugated to FITC. We used IgG1 (Immunotech) as negative controls. Stained cells were acquired on a FACSAria^TM^ (Becton Dickinson, Franklin Lakes, NJ, USA) flow cytometer and analyzed using Flow Jo v10.0.7 software (FlowJo LLC, Ashland, OR, USA).

### 4.6. Histological Analysis.

Insulitis was evaluated on snap-frozen acetone-fixed cryosections (8-µm thick). Hematoxylin and eosin staining (Thermo Electron Corp, Waltham MA, USA) were performed, and we evaluated the degree of inflammation microscopically. The percentage of non-infiltrated, peri-infiltrated, slightly infiltrated (less than 50% of islet area), or highly infiltrated (more than 50% of islet area) islets were evaluated.

### 4.7. Bone-Marrow Derived DCs

DCs were derived from bone marrow culture (BM-derived DCs) from NOD and NOR mice, as described [10]. At day 6, non-adherent cells were harvested and a CD11c^+^ sorting was performed using magnetic beads (purity > 95%) (Miltenyi Biotech, Bergisch Gladbach, Germany). CD11c^+^ cells were treated with different concentrations of CoPP (Frontier Scientific, Carnforth, UK), as previously described [39]

### 4.8. Isolation of Splenic DCs

Single-cell suspensions were prepared by enzymatic spleen disaggregation with collagenase D (Sigma–Aldrich) and CD11c^+^ population was enriched by cell sorting using magnetic beads (purity > 95%) (Miltenyi Biotech). The DCs from conditional transgenic mice were incubated for 24 h with 0.5 or 2 μg/mL of doxycycline.

### 4.9. Statistics

For diabetes incidence, significance was calculated using the log-rank test. For all other parameters, significance was calculated by *t*-paired test, Mann–Whitney non-parametric *t*-test. One-way ANOVA or two-way ANOVA using Prism software (GraphPad Software, San Diego, CA, USA): * *p* < 0.05, ** *p* < 0.01, and *** *p* < 0.001.

## Figures and Tables

**Figure 1 ijms-20-01676-f001:**
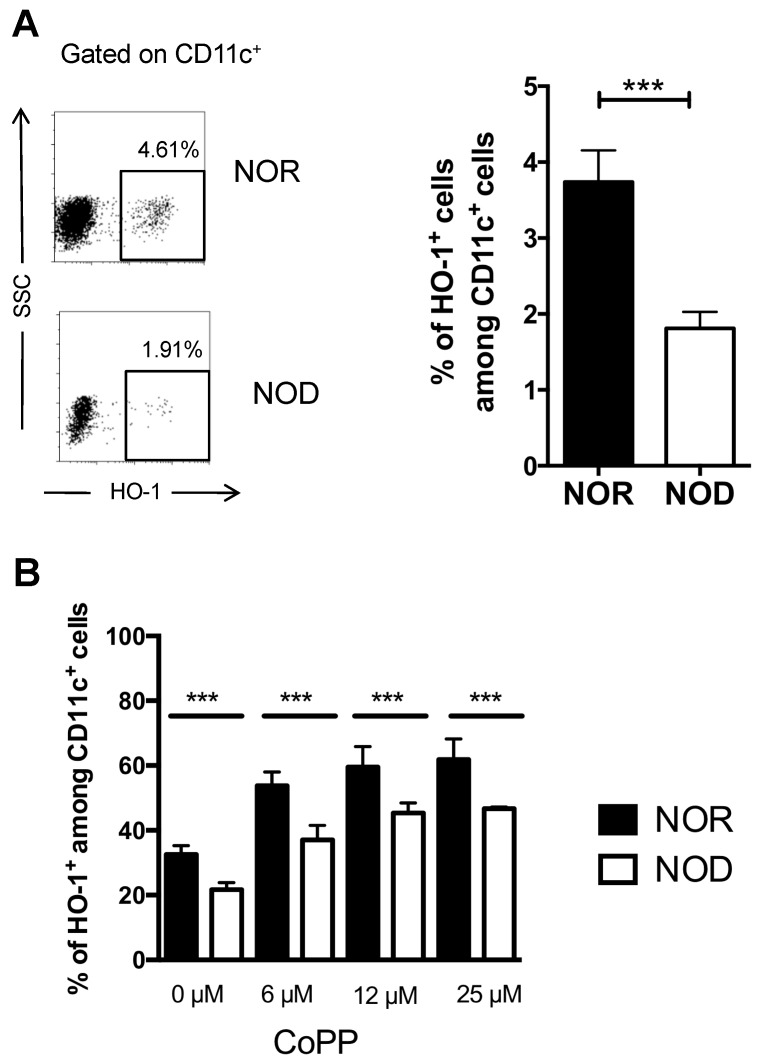
Heme oxygenase-1 expression deficiency in NOD DCs. (**A**) HO-1 expression in splenic CD11c^+^ cells was evaluated in one-month-old female NOD and NOR mice by flow cytometry. Representative dot plots (left panel) and mean frequencies of splenic HO-1^+^ CD11c^+^ cells ± s.e.m. (*n* ≥ 7 mice/group) (right panel) are reported. (**B**) Flow cytometry analysis showing HO-1 expression in CD11c^+^ of bone marrow-derived DCs (BMDC) from NOD and NOR mice treated or not with CoPP for 24 h. Mean frequencies of splenic HO-1^+^ CD11c^+^ cells ± s.e.m. (*n* ≥ 8 mice/group) from 3 independent experiments are reported. An unpaired *t*-test (**A**) and a one-way ANOVA followed by Tukey’s post-hoc test (**B**) were performed. ***, *p* < 0.001.

**Figure 2 ijms-20-01676-f002:**
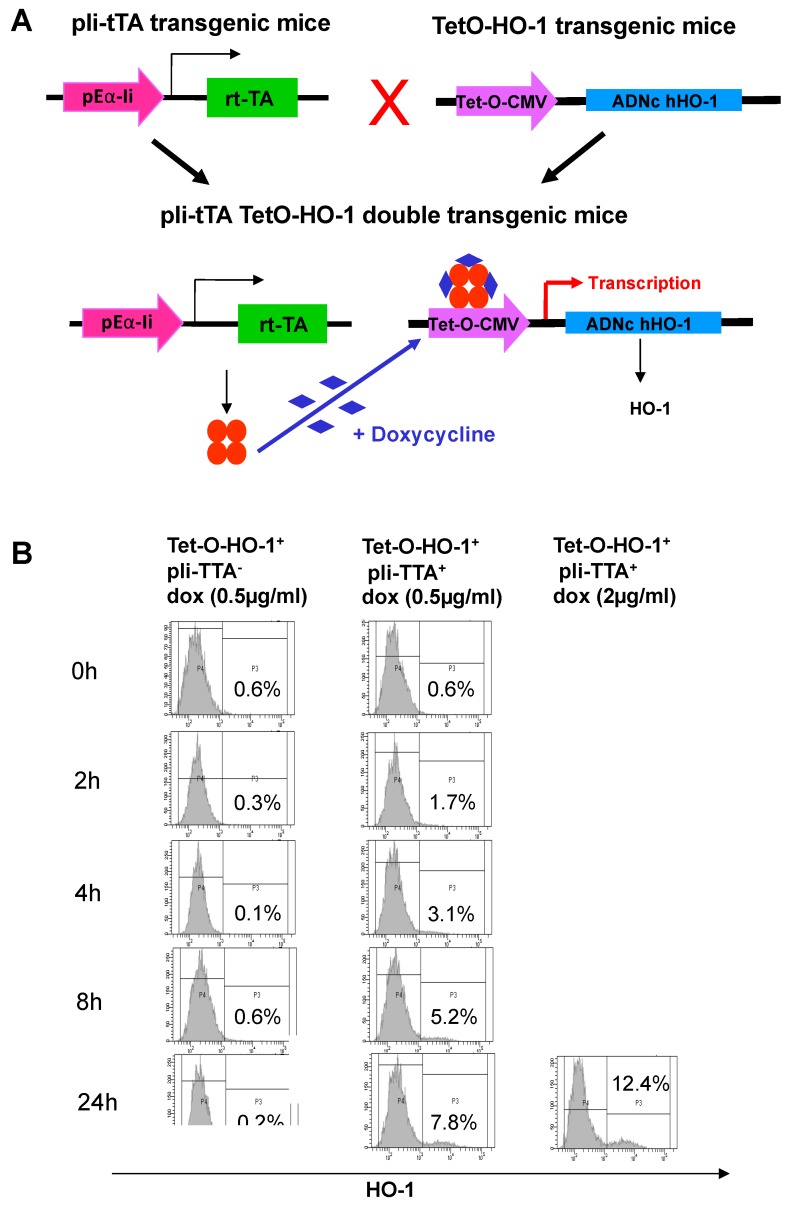
In vitro doxycycline treatment of splenic CD11c^+^ DCs from double transgenic TetO-HO-1^+^ pIi-tTA^+^ NOD mice induces HO-1 expression. (**A**) Schematic representation of the transgenic mouse strains used in the study. pIi-tTA are transgenic mice in which the tetracycline transactivator (*tTA*) gene is under the control of the Major Histocompatibility Complex II (MHC-II) invariant chain (Eα-Ii) promoter [24]. TetO-HO-1 are transgenic mice in which the human *homx-1* cDNA was cloned downstream a hybrid cytomegalovirus (CMV)-Tet operator rtTA. Double transgenic mouse strain show induction of HO-1 expression in MHC-II^+^ cells in the presence of doxycycline (DOX). (**B**) Splenic CD11c^+^ DCs from single transgenic Tet-O-HO-1^+^pli-tTA^−^ or double transgenic TetO-HO-1^+^ pIi-tTA^+^ NOD mice were cultured with doxycycline for 24 h and analyzed by flow cytometry for HO-1 expression. One representative experiment out of two is shown.

**Figure 3 ijms-20-01676-f003:**
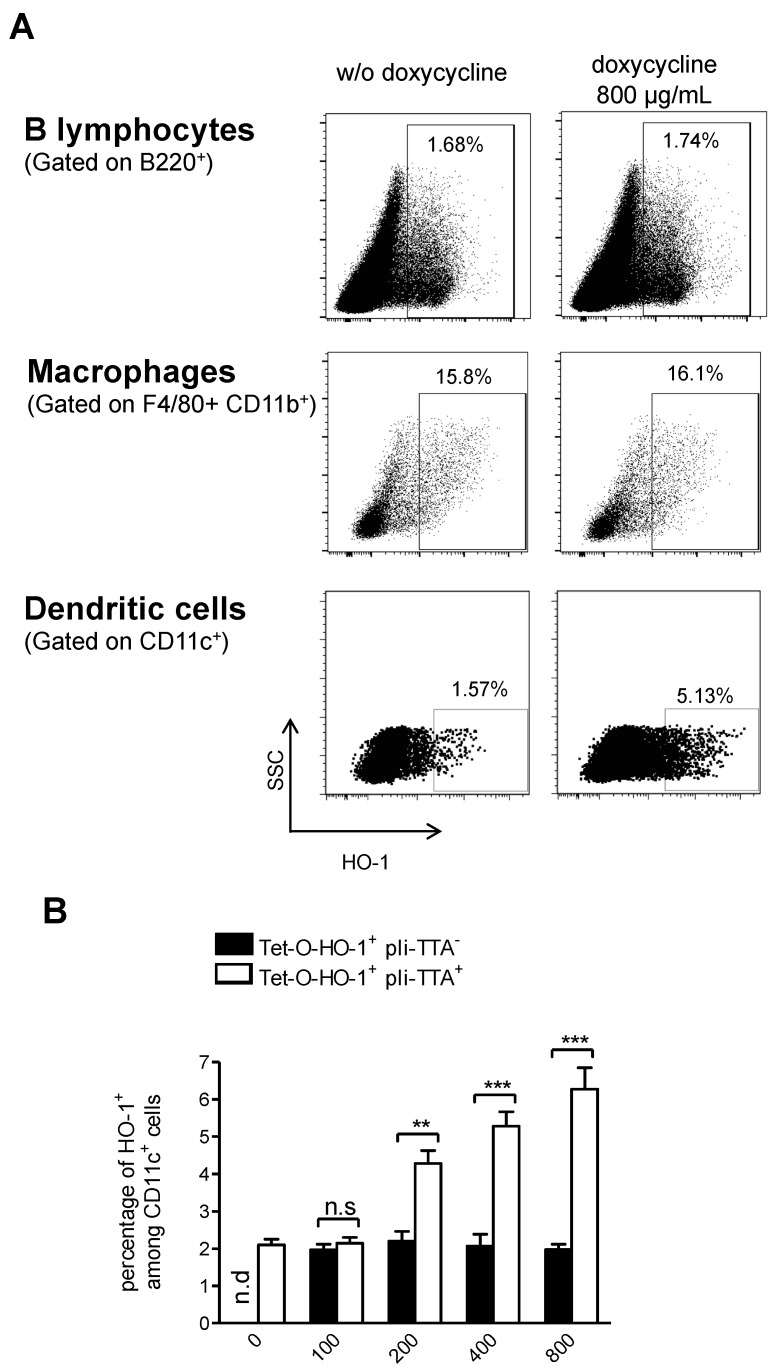
Induction of HO-1 expression by addition of doxycycline in the drinking water of double transgenic TetO-HO-1^+^ pIi-tTA^+^ NOD mice. (**A**,**B**) Different doses of doxycycline (DOX) were added to the drinking water in single transgenic Tet-O-HO-1^+^pli-tTA^−^ or double transgenic TetO-HO-1^+^ pIi-tTA^+^ NOD mice to induce HO-1 expression. (**A**) Splenic cell populations (DCs cells, B lymphocyte or macrophages) were analyzed for HO-1 expression by intracellular HO-1 staining. (**B**) Expression of HO-1 in splenic DCs are presented as mean frequencies ± s.e.m. (*n* ≥ 4 mice/group) for two independent experiments. One-way ANOVA followed by Tukey’s post-hoc tests were performed. *** *p* < 0.001. n.s.: not significant; n.d.: not determined.

**Figure 4 ijms-20-01676-f004:**
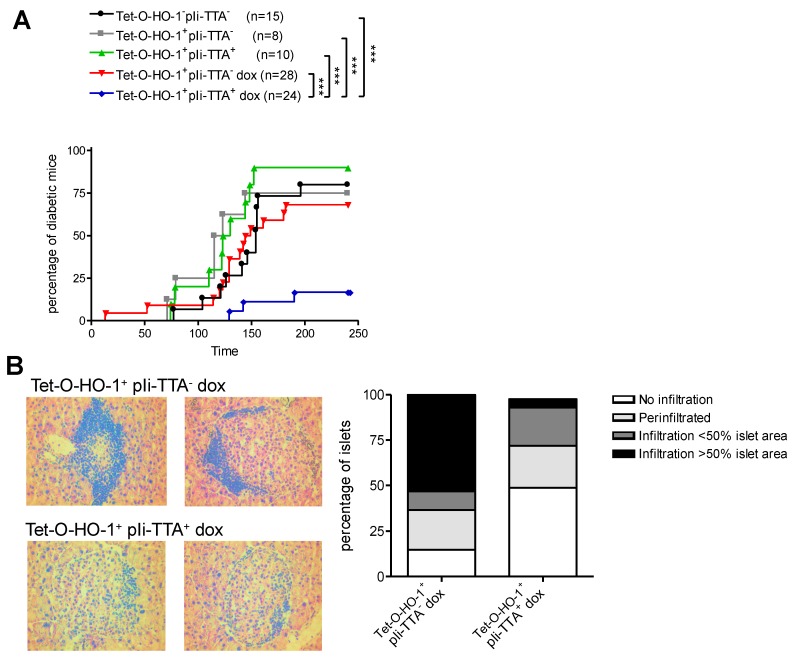
HO-1 induction in DCs decreased diabetes incidence in NOD mice. (**A**,**B**) Starting at one month of age, non-transgenic, single transgenic Tet-O-HO-1^+^pli-tTA^−^ or double transgenic TetO-HO-1^+^ pIi-tTA^+^ female NOD mice were provided or not with 800 µg/mL of doxycycline (DOX) in their drinking water. (**A**) Cohorts were monitored for the development of diabetes by measurement of glycemia. Mice were considered diabetic when glycemia was over 200 mg/dL for two consecutive days. (**B**) In three-month-old female single transgenic Tet-O-HO-1^+^pli-tTA^−^ or double transgenic TetO-HO-1^+^ pIi-tTA^+^ NOD mice, insulitis was evaluated by hematoxylin and eosin (H&E) staining of pancreatic sections. Representative H&E staining of pancreatic sections are shown (left panels). The extension of insulitis is reported (right panel) as the percentage of non-infiltrated, peri-infiltrated, slightly infiltrated (less than 50% of islet area), or highly infiltrated (more than 50% of islet area) islets (*n* > 4 mice/group). The total numbers of analyzed islets for single transgenic Tet-O-HO-1^+^pli-tTA^−^ or double transgenic TetO-HO-1^+^ pIi-tTA^+^ female transgenic NOD mice were 52 and 47, respectively. (**A**) Log-rank test was performed, and only significant differences were reported. *** *p* < 0.001.

**Figure 5 ijms-20-01676-f005:**
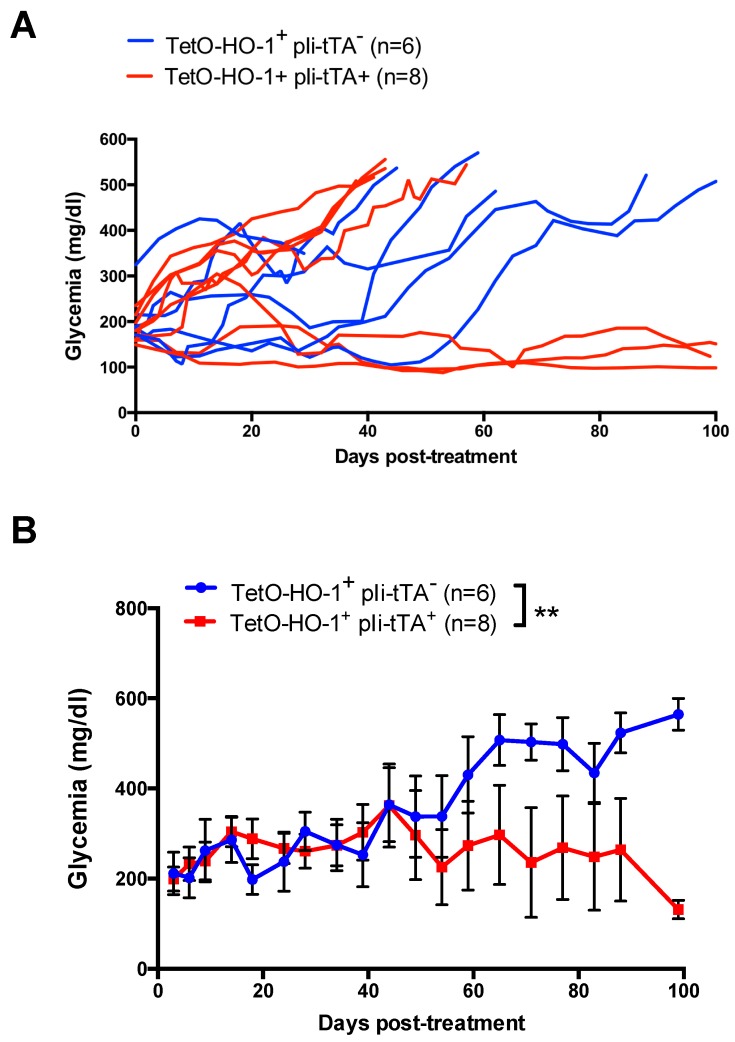
HO-1 induction in DCs inhibited diabetes in spontaneously diabetic mice. Diabetic mice with a glycemia between 200 mg/dL and 250 mg/dL for two consecutive days were identified in single transgenic Tet-O-HO-1^+^pli-tTA^−^ or double transgenic TetO-HO-1^+^ pIi-tTA^+^ female NOD mice cohorts. These diabetic mice were treated with doxycycline (DOX) in the drinking water (800 µg/mL). Mice then were monitored for glycemia twice a week. Graphs show glycemia over time for each animal (**A**) or as a mean ± s.e.m. (*n* ≥ 6 mice/group) (**B**). Log-rank test was performed in (**B**). ** *p* < 0.01.

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
