# Peer review of "Genetic Restoration of Heme Oxygenase-1 Expression Protects from Type 1 Diabetes in NOD Mice"

_ijms, 2019, doi:10.3390/ijms20071676_

Round 1

Reviewer 1 Report

The present manuscript entitled “Genetic restoration of heme-oxygenase-1 expression protects from type 1 diabetes in NOD mice” by J. Pogu et al., pointed out Mice in which HO‐1 expression was induced in DCs exhibited a lower T1D incidence and a reduced insulitis compared to non‐induced mice. Further authors found upregulation of HO‐1 in DCs prevent increase of glycemia in diabetic NOD mice. Based overall this study remarked that induction of HO‐1 expression in DCs as a preventative treatment for T1D. Overall, the exhibited work is novel, elegantly composed, and the article is all around abridged. However, introduction lines (32-35, the authors pointed out that HO‐1 is the only one which is induced by oxidative stress or proinflammatory cytokines. It is also induced by its own substrates, whether they are natural (heme or hemin) or synthetic (Cobalt‐protoporphyrin, CoPP). This is interesting but to strengthen this study, if author could discuss the how oxidative stress plays a role in activation of HO-1 more elaborately based on recently published Papers.

Author Response

We thanks this reviewer for positive comments

In the revised version of the paper we have improved the introduction by mentioning the induction of HO-1 by oxidative stress and we are now discussing how this could contribute to the inhibition of T1D observed after induction of HO-1 in DCs.

Reviewer 2 Report

The manuscript is of high relevance and interest. The ability of immune cells to cope with oxidative stress and/or its derivatives is an underexplored topic in immunity research. I support the publication of the work, provided the some minor points will be adequately addressed:

Results

- reference to "Fig. 1A" is missing

- was the percentage of HO-1 positive b cells and macrophages in NOD splenocytes (Fig. 1a) assessed? data as supplement would be fine.

- IHC images should be added to appendix to support data on onset of diabetes in figure 4b

- figure legend 5: reference to B and A should be vice versa?

- figure 5a should be be made more clear. one idea would be to extend the y-scale (no clipping as seen in the moment) and to use smooting tools in prism to show more the general trend

- provide a supplemental file showing the gating strategy to obtain your populations shown in figure 3

discussion

- what is the relevance of only a few dcs being positive for HO-1? what do you speculate the other 95% are doing?

- why is it that still 5 of 8 might are dying in TA+ animals in figure 5? what are alternative routes or possible combination treatment that would increase this survival 

- specifiy the molecules given when refering to  32 and 33

- attenuate your conclusion drawn from b cell and macrophages in line 207: there is no quantification of b cells and macrophages but only dot plots, which are ok but not highly convincing; same applies to your DC population cut-off at the side scatter; viability dye is also missing in the panel (e.g. fixable viabilities dyes); gating is semi up to date, see yu et al. 2016 plosone

- discuss a bit on the antioxidant properties of HO-1 and how that may contribute to the effects observed by you

M/M

- 4.8 "sigma-aldricht" without the "t"

Author Response

We thanks this reviewer for positive comments.

The manuscript is of high relevance and interest. The ability of immune cells to cope with oxidative stress and/or its derivatives is an underexplored topic in immunity research. I support the publication of the work, provided the some minor points will be adequately addressed:

Results

- reference to "Fig. 1A" is missing. We have added the reference missing (line 77).

- was the percentage of HO-1 positive b cells and macrophages in NOD splenocytes (Fig. 1a) assessed? data as supplement would be fine. We agree with the reviewer that adding this data to the manuscript would have been nice. Unfortunately we didn’t assessed the expression of HO-1 neither in macrophage nor in B lymphocyte in this study. So we cannot rule out that there is other differences in HO-1 expression on other cell types than DCs. We have added a sentence on this topic in the discussion (line 201-203).

- IHC images should be added to appendix to support data on onset of diabetes in figure 4b. These data have been added in figure 4B.

- figure legend 5: reference to B and A should be vice versa? We have modified the text to correct this mistake (line 188).

- figure 5a should be made more clear. one idea would be to extend the y-scale (no clipping as seen in the moment) and to use smoothing tools in prism to show more the general trend. We have extended the y-scale and used the smoothing tool as suggested.  

- provide a supplemental file showing the gating strategy to obtain your populations shown in figure 3. We are showing the gating strategy in the revised version in supplemental figure 1.

discussion

- what is the relevance of only a few dcs being positive for HO-1? what do you speculate the other 95% are doing? We have added a comment on this point in the discussion (line 229).

- why is it that still 5 of 8 might are dying in TA+ animals in figure 5? what are alternative routes or possible combination treatment that would increase this survival. We have also added a comment on this point in the discussion (line 232-235).

- specify the molecules given when referring to  32 and 33. This point have been added to the revised version (line 210, 211).

- attenuate your conclusion drawn from b cell and macrophages in line 207: there is no quantification of b cells and macrophages but only dot plots, which are ok but not highly convincing; same applies to your DC population cut-off at the side scatter; viability dye is also missing in the panel (e.g. fixable viabilities dyes); gating is semi up to date, see yu et al. 2016 plosone. We agree with the reviewer and have moderated our conclusion as requested (line 218).

- discuss a bit on the antioxidant properties of HO-1 and how that may contribute to the effects observed by you. We have also added a comment on this point in the discussion (line 254-260).

M/M

- 4.8 "sigma-aldricht" without the "t". Corrected.